# Severity of non-alcoholic steatohepatitis is not linked to testosterone concentration in patients with type 2 diabetes

Kristin Alexandra Dayton[1], Fernando Bril[2,3], Diana Barb[2], Jinping Lai[4], Srilaxmi Kalavalapalli[2], Kenneth Cusi[2,3]*

1 Department of Pediatrics, Division of Pediatric Endocrinology, University of Florida, Gainesville, Florida, United States of America, 2 Department of Medicine, Division of Endocrinology, Diabetes and Metabolism, University of Florida, Gainesville, Florida, United States of America, 3 Malcom Randall Veterans Administration Medical Center, Gainesville, Florida, United States of America, 4 Department of Pathology, Immunology and Laboratory Medicine, University of Florida, Gainesville, Florida, United States of America

* Kenneth.Cusi@medicine.ufl.edu

## Abstract

**Data Availability Statement:** All relevant data are within the paper and its Supporting information files.

### Background

Hypogonadism is reported to occur in non-alcoholic fatty liver disease (NAFLD), but earlier studies used low-sensitivity diagnostic techniques (CT, ultrasound), for NAFLD diagnosis. We hypothesized that if hypogonadism was due to NAFLD, and not solely attributable to underlying obesity/diabetes, it would be more severe in the presence of steatohepatitis (NASH). To examine the influence of liver disease on testosterone in males with type 2 diabetes mellitus (T2DM), we used gold-standard liver imaging with MR-spectroscopy ($^1$H-MRS), and performed liver biopsies to grade/stage the NAFLD.

### Methods

In this cross-sectional study, we measured in 175 males with T2DM total and free testosterone, markers of insulin resistance, and intrahepatic triglyceride content (IHTG) by $^1$H-MRS. Those with NAFLD on imaging underwent a liver biopsy.

### Results

Total testosterone was higher in the group without NAFLD ("No-NAFLD"; n = 48) compared to isolated steatosis (IS; n = 62) or NASH (n = 65) (385 ± 116 vs. 339 ± 143 vs. 335 ± 127 ng/ml, $p_{trend}$ 0.03). Testosterone was also lower in obese vs. non-obese subjects in both the No-NAFLD and IS groups (p = 0.06 and p = 0.11, respectively), but not in obese vs. non-obese patients with NASH (p = 0.81). IHTG was independently associated with total testosterone (ß = -4.8, p = 0.004). None of the liver histology characteristics were associated with lower testosterone.

**Funding:** This study was partially funded by the American Diabetes Association (1-08-CR-08 [K. C.]) https://www.diabetes.org/ and the U.S. Department of Veterans Affairs, VA Merit Award (1 I01 CX000167-01 [K. C.]), https://www.research. va.gov/funding/. The funders had no role in study design, data collection and analysis, decision to publish, or preparation of the manuscript.

## Conclusions

NAFLD is linked to lower total testosterone in patients with T2DM, but likely given a common soil of insulin resistance/obesity and not from the severity of liver necroinflammation or fibrosis. Nevertheless, clinicians should consider screening patients with T2DM and NAFLD for hypogonadism.

## Introduction

Nonalcoholic fatty liver disease (NAFLD) encompasses a spectrum of disease that ranges from isolated hepatic steatosis to its more severe form with hepatocyte ballooning, lobular inflammation and often fibrosis, also known as non-alcoholic steatohepatitis (NASH). NAFLD is estimated to affect at least 25% of the adult population [1] and will soon be the most common indication for liver transplantation in the United States [1–3]. In the type 2 diabetes mellitus (T2DM) population, NAFLD affects 60–70% of patients [4, 5] with 50–70% of those patients demonstrating changes consistent with NASH [6, 7].

Acquired hypogonadism is commonly associated with obesity and metabolic syndrome in males [8]. Studies have also shown a strong association of T2DM and hypogonadotropic hypogonadism in males, with strong correlations of testosterone levels with obesity and insulin resistance in these studies as well [9, 10]. Though the mechanism for this is largely unknown, it is hypothesized that increased aromatization of androgens in adipose tissue to estrogens leads to feedback inhibition of gonadotropins, leading to hypogonadotropic hypogonadism through reduced testosterone synthesis in the testes [11]. Additionally, increased inflammatory cytokines from adipose tissue and elevated leptin levels in association with central nervous system leptin resistance and insulin resistance may also contribute to decreased testosterone production in obesity and T2DM [12, 13].

Nonalcoholic fatty liver disease has been strongly associated with obesity, T2DM and the metabolic syndrome [7]. The impact of having NAFLD, or even more severe liver disease, such as NASH, in terms of risk for hypogonadism remains uncertain [14]. This is important as it has significant clinical management implications. Several studies have investigated this association with mixed results. While many of these have shown an association of NAFLD with low testosterone levels, they have relied on low-sensitivity tools for the diagnosis of NAFLD, such as liver ultrasound or computed tomography (CT) [15–19]. A study by Seo and colleagues showed that NAFLD may be associated with low testosterone, but that this effect was not independent of underlying obesity and metabolic risk factors [20]. To the best of our knowledge, no prior studies have used the gold-standard (liver histology) for the diagnosis of NAFLD/NASH in this setting, to fully establish if there is an independent association between NAFLD severity and hypogonadism. To this, we aimed to investigate in-depth, using gold-standard testing, the association of testosterone levels with NAFLD and NASH in patients with T2DM.

## Methods

### Subjects

Approval was obtained from the Institutional Review Boards of University of Texas H.S.C at San Antonio, Texas and the University of Florida at Gainesville, FL. Written consent was obtained from all study participants. A total of 175 male patients with a diagnosis of T2DM were recruited from the general population or hepatology and endocrinology clinics in

Gainesville, Florida and San Antonio, Texas. Metformin, sulfonylureas and insulin were the only anti-hyperglycemic medications allowed. For calculations of insulin resistance (HOMA-IR, Matsuda Index and Adipo-IR), patients on insulin were excluded for the analyses. Patients were excluded if they were on pioglitazone, GLP-1 agonists, vitamin E, or any weight loss medication. Patients on testosterone replacement were also excluded from the study. Other exclusion criteria included: significant alcohol consumption ($\geq$30g/day for men and $\geq$20 g/day for women) or any liver disease other than NASH (hepatitis B or C, autoimmune hepatitis, hemochromatosis, Wilson's disease or drug-induced hepatitis). Patients were also excluded if they had type 1 diabetes mellitus or any evidence of clinically significant renal, pulmonary or heart disease.

### Study design

In this cross-sectional study, all patients underwent routine laboratory tests, including fasting total and free testosterone levels. Patients underwent proton magnetic resonance spectroscopy ([1]H-MRS) to measure intrahepatic triglyceride content, and a 75-g oral glucose tolerance test to assess insulin secretion and action. Patients with a diagnosis of NAFLD based on [1]H-MRS were offered a percutaneous liver biopsy to establish the diagnosis of NASH and to grade and stage the disease.

### Measurements of intrahepatic triglyceride content

Measurement of intrahepatic triglyceride content was performed by [1]H-MRS. Two or three areas with a volume of 30 × 30 × 30 mm were selected for voxel placement within the right lobe of the liver. A single experienced observer analyzed the spectra using commercial software (NUTS; Acorn NMR, Inc., Livermore, California). Intrahepatic triglyceride content was calculated as fat fraction (area under the curve [AUC] fat peak/[AUC fat peak + water peak]). Measurements were corrected for T1 and T2 relaxation using methods previously described. A liver fat content of >5.56% was considered diagnostic of NAFLD.

### Analytical measurements

Total testosterone levels (normal range 250–1100 ng/mL) were measured by electro-chemiluminescence immunoassay (Cobas 602, Roche Diagnostics International Ltd, Switzerland) and free testosterone (normal range 35–155 pg/mL) by liquid chromatography tandem mass spectroscopy (Quest Diagnostics Nichols Institute, Valencia, CA). Hypogonadism was defined as total testosterone less than 250 ng/mL.

### Percutaneous liver biopsy

An ultrasonography-guided liver biopsy was performed in patients with a diagnosis of NAFLD by [1]H-MRS. Histological characteristics for the diagnosis of NASH were determined using standard criteria [21]. Briefly, a diagnosis of definite NASH was made based on the presence of: zone 3 accentuation of macrovesicular steatosis (any grade), hepatocellular ballooning (of any degree) and lobular inflammatory infiltrates (of any amount). The NAFLD activity score was calculated as the sum of the steatosis, inflammation and ballooning grades in the liver biopsy.

### Statistical analysis

Data was expressed as percentages for categorical variables and as mean ± standard deviation for numerical variables (except when noted). Comparisons among categorical variables were

performed by $\chi^2$ or Fisher Exact test. Kruskal-Wallis or oneway ANOVA were used for comparisons between numerical variables, depending on their distribution (post-hoc pairwise comparisons were performed by Steel-Dwass and Tukey-Kramer, respectively). Pearson's correlation and multiple linear regression analyses was also used to assess the association between numerical variables. A two-tailed P < 0.05 was considered statistically significant. All analysis was performed using JMP Pro 13 (SAS Institute Inc., Cary, NC, USA) and Stata 11.1 (StataCorp LP).

## Results

### Population characteristics

Characteristics of the population are summarized in Table 1. As can be observed, although age was statistically different among groups, this difference was not clinically relevant (57 vs. 60 vs. 63 years). Groups were well-matched for diabetes control (both fasting plasma glucose and

**Table 1. Population characteristics by liver histology.**

| | No NAFLD (n = 48) | Isolated Steatosis (n = 62) | NASH (n = 65) | P-value |
|---|---|---|---|---|
| **Age, years** | 63 (7) | 60 (8) | 57 (9)* | **<0.001** |
| **Ethnicity** | | | | **<0.001** |
| Caucasian, % | 69% | 76% | 59% | |
| Hispanic, % | 8% | 14% | 36% | |
| African American, % | 23% | 10% | 5% | |
| **BMI, kg/m2** | 31.4 (4.5) | 33.7 (4.7)* | 34.5 (4.3)* | **0.002** |
| **Fasting plasma glucose, mg/dl** | 155 (52) | 154 (38) | 147 (36) | 0.46 |
| **HbA1c, %** | 7.1 (1.2) | 7.0 (1.1) | 7.5 (1.4) | 0.099 |
| **Fasting plasma insulin, uU/ml** | 8.9 (6.2) | 14.7 (11.2)* | 21.1 (14.4)*# | **<0.001** |
| **Use of diabetes medications** | | | | |
| Metformin, % | 84% | 83% | 70% | 0.15 |
| Sulfonylurea, % | 46% | 38% | 45% | 0.73 |
| Insulin, % | 39% | 19% | 24% | 0.16 |
| **Cholesterol, mg/dl** | 150 (24) | 166 (37) | 174 (45) | 0.13 |
| **LDL, mg/dl** | 83 (23) | 92 (33) | 94 (34) | 0.37 |
| **HDL, mg/dl** | 43 (9) | 40 (12) | 36 (9) | 0.057 |
| **Liver fat content, %** | 2.5 (1.4) | 10.7 (4.9)* | 16.7 (6.9)*# | **<0.001** |
| **ALT, IU/L** | 23 (8) | 36 (24) | 76 (40)*# | **<0.001** |
| **AST, IU/L** | 22 (5) | 27 (13) | 52 (25)*# | **<0.001** |
| **Albumin, g/L** | 4.5 (0.2) | 4.4 (0.3) | 4.3 (0.4) | **0.001** |
| **HOMA-IR** | 2.9 (2.0) | 5.9 (5.4)* | 6.9 (5.3)* | **0.002** |
| **Adipo-IR, mmol/L. uU/ml** | 2.7 (1.4) | 5.4 (4.3) | 8.8 (6.8)*# | **<0.001** |
| **Matsuda Index** | 4.6 (4.0) | 3.3 (2.2)* | 1.9 (0.8)* | **<0.001** |
| **NAFLD Activity Score** | | 2.7 (1.4) | 4.8 (1.1) | **<0.001** |
| Steatosis grade | | 1.3 (0.7) | 1.9 (0.7) | **<0.001** |
| Inflammation grade | | 1.4 (0.6) | 1.9 (0.3) | **<0.001** |
| Ballooning grade | | 0.0 (0.0) | 0.9 (0.4) | **<0.001** |
| Fibrosis stage | | 0.5 (0.6) | 1.7 (1.3) | **<0.001** |

Data are expressed as mean (SD) unless otherwise noted.

* = p<0.05 for difference from No NAFLD,

# = p <0.05 for difference from isolated steatosis.

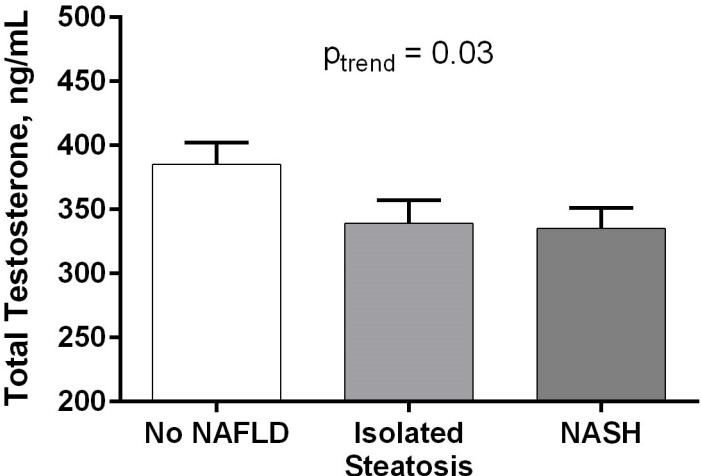

**Fig 1. Total and free testosterone levels by liver histology in subjects with type 2 diabetes.** Data expressed as mean ± SEM. After adjustment for multiple comparisons, between-group comparisons in Fig 1 are Group 1 vs. 2: p = 0.19, Group 2 vs. 3: p = 0.99, Group 1 vs. 3: p = 0.13.

HbA1c) and for anti-hyperglycemic medication use. BMI was significantly higher in patients with NAFLD (isolated steatosis or NASH) compared to patients without NAFLD. Of note, we observed no difference in weight or BMI between patients with isolated steatosis (IS) and those with NASH. In the group without NAFLD, 42% were non-obese while 58% were obese, in patients with isolated steatosis 24% were non-obese while 76% were obese, and in the group with NASH 20% were non-obese while 80% were obese. Fasting plasma insulin and fasting insulin resistance measurements were significantly higher in patients with NAFLD, and more so in those with NASH. The Matsuda index was significantly lower in patients with IS and NASH compared to those without NAFLD. Those patients with steatosis and NASH had higher AST and ALT levels with more severe histologic findings.

## Total and free testosterone levels across the spectrum of NAFLD

As observed in Fig 1, total testosterone levels were higher in those without NAFLD ("No NAFLD") compared to patients with IS and NASH (385 ± 116 vs. 339 ± 143 vs. 335 ± 127 ng/ml, p for trend 0.03), but not the free testosterone concentration ($p_{trend}$ = 0.97). We observed a significantly lower proportion of patients with hypogonadism (defined as total testosterone <250 ng/ml) in the No NAFLD group compared to IS and NASH groups (8.3% vs. 32.3% vs. 27.7%, p = 0.01), but not different when based on free testosterone levels (<35 pg/ml).

In order to determine whether the association between testosterone levels and severity of liver disease was independent of obesity, patients were stratified by presence or absence of obesity. The results of this analysis are represented in Fig 2. As can be observed, in patients without NAFLD and those with isolated steatosis, total testosterone levels were higher in non-obese vs. obese subjects (p = 0.06 and p = 0.11, respectively). However, there were no differences in total testosterone between non-obese and obese patients in the NASH group (p = 0.81). A similar trend was observed with plasma free testosterone levels among the groups comparing non-obese to obese individuals (No NAFLD: 63.7 ± 5.8 vs. 56.2 ± 3.8 pg/ml, p = 0.27; Isolated steatosis: 66.6 ± 8.7 vs. 54.7 ± 3.5 pg/ml, p = 0.14; and NASH: 60.4 ± 4.7 vs. 59.4 ± 3.6 pg/ml, p = 0.89).

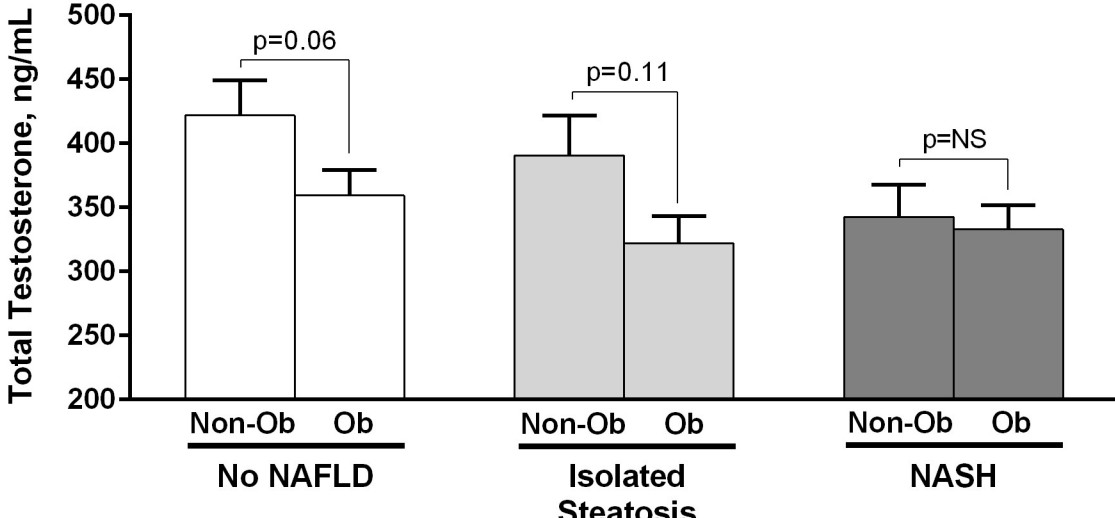

**Fig 2. Total testosterone levels by liver histology in non-obese (No) vs obese (Ob) subjects with Type 2 diabetes.** Data expressed as mean ± SEM.

To further investigate this association of liver fat with total and free testosterone levels, we analyzed the correlations between testosterone levels and intrahepatic triglyceride content measured by ¹H-MRS. As seen in Fig 3B, there is a downward trend in testosterone levels as degree of hepatic steatosis increases. A similar downward trend in testosterone was also seen with increasing BMI and worsening insulin resistance as measured by HOMA-IR (Fig 3A and 3C). In order to assess whether the association between total plasma testosterone and intrahepatic triglyceride content was independent of other metabolic factors, we performed a multiple linear regression analysis (Table 2). Intrahepatic triglyceride content was independently associated with total testosterone levels (ß = -4.8, p = 0.004). Body mass index (ß = -9.6, p<0.001) and HbA1c (ß = -19.8, p = 0.02) were also independently associated with total testosterone in the multiple linear regression.

In order to determine the relationship between the severity of liver histology and testosterone levels, we analyzed total testosterone levels by each individual histological parameter. As described in Fig 4, there were no significant differences in total testosterone levels by steatosis,

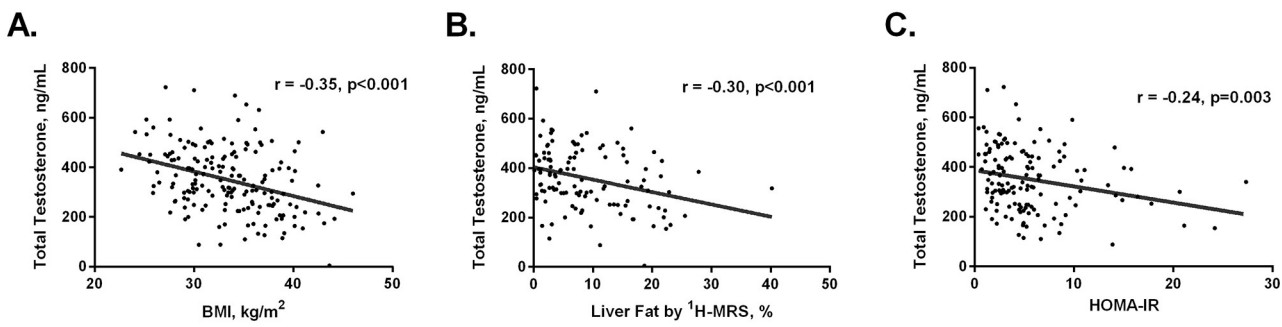

**Fig 3. Total testosterone levels by BMI (A), liver fat on ¹H-MRS (B) and HOMA-IR (C).**

**Table 2. Results of multiple linear regression analysis for variables associated with total testosterone.**

|  | Coefficient | p | 95% confidence interval | |
|---|---|---|---|---|
| Liver fat content | -4.8 | 0.004 | -7.9 | -1.6 |
| HbA1c | -19.8 | 0.024 | -37.0 | -2.7 |
| BMI | -9.6 | <0.001 | -13.8 | -5.3 |
| ALT | 0.8 | 0.016 | 0.2 | 1.5 |

HDL, TG, Age, FPG, AST were also tested but were not independently associated with total testosterone.

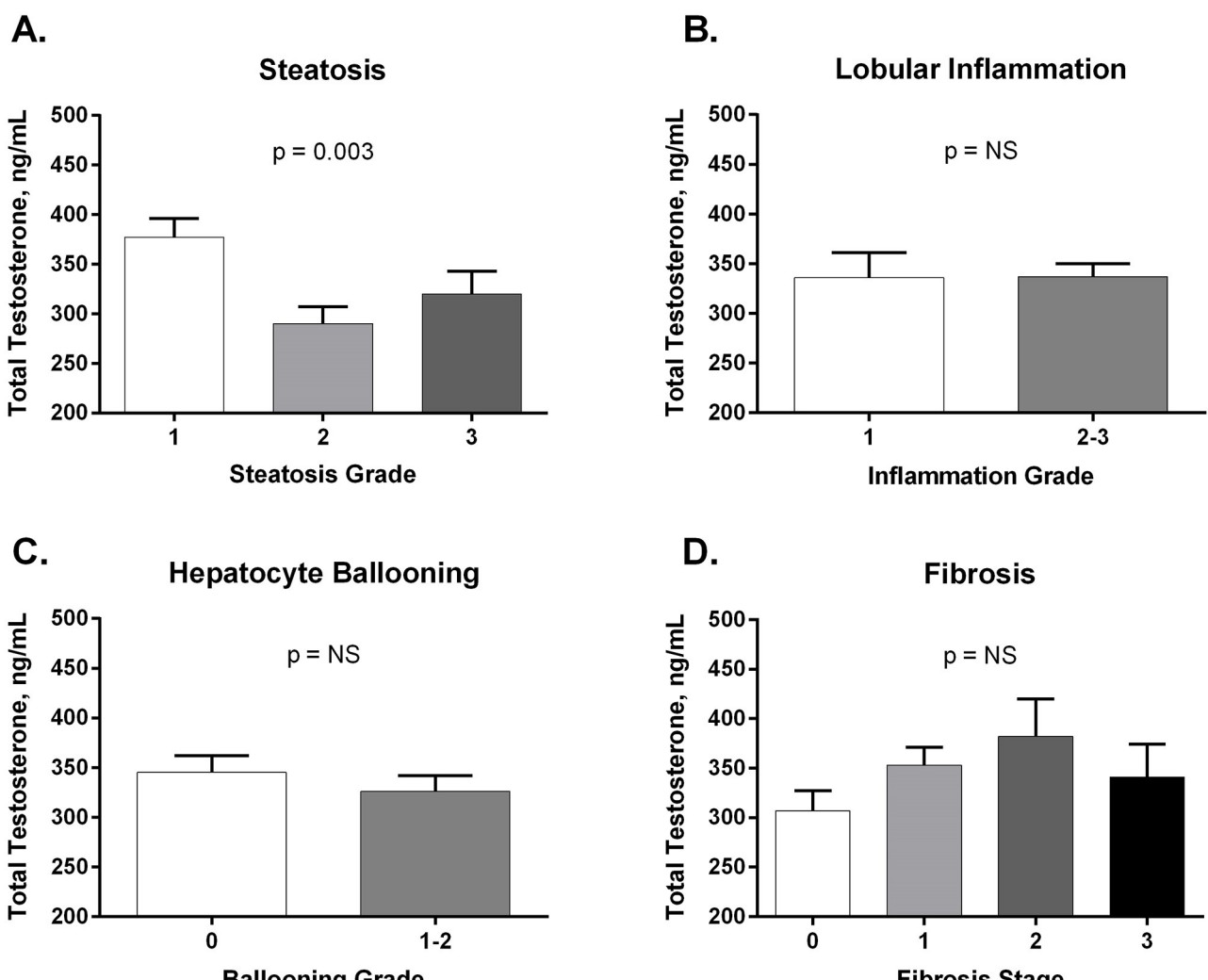

**Fig 4. Analysis of total testosterone levels by histology characteristics (panels A-D).** Data expressed as mean ± SEM. Fig 4A. After adjustment for multiple comparisons, between-group comparisons are Steatosis grade 1 vs. 2: p = 0.003, Steatosis grade 2 vs. 3: p = 0.99, Steatosis grade 1 vs. 3: p = 0.37. Fig 4D. After adjustment for multiple comparisons, between-stage groups comparisons are Stage 0 vs. 1: 0.56, Stage 1 vs. 2: 0.99, Stage 0 vs. 2: 0.65, Stage 0 vs. 3: 0.99, Stage 2 vs. 3: 0.99.

inflammation, or ballooning grade (panels A-C), or fibrosis stage (panel D). A similar pattern was observed with free testosterone levels.

## Discussion

There is ongoing debate regarding the relationship of NAFLD to hypogonadism. Previous research has shown an association of NAFLD with hypogonadism [14, 22]. However, studies have been conflicting regarding whether these associations are due solely to the confounding factor of obesity, or if there may be an underlying role of liver fat and inflammation on testosterone levels independent of the effects of obesity [14, 22]. While several studies have attempted to answer this question, they have relied on lower sensitivity techniques for NAFLD diagnosis. Our study was unique in its use of biopsy-proven NASH to evaluate for the underlying relationship between NAFLD and testosterone levels. We found no association between testosterone levels and any of the liver histology characteristics of NASH (inflammation, ballooning or fibrosis). This crucial finding adds to our knowledge of the underlying mechanisms for testosterone deficiency in patients with NAFLD. As increasing NASH severity was not associated with lower testosterone levels, it follows that another factor is driving the progressive hypogonadism as hepatic steatosis worsens. The most likely relationship lies in the link between hepatic steatosis and insulin resistance, and in concordance with this, there was a close association of hepatic steatosis with insulin resistance in our study.

Furthermore, there is also a question of whether obesity alone may be the driver for hypogonadism in patients with NAFLD. Previous studies found that insulin resistance may play a role in hypogonadism, with conflicting evidence on whether obesity is the underlying driver of this relationship [23, 24]. Our results demonstrated that liver fat serves as a strong predictor of total testosterone levels, and this effect occurs independent of the presence of obesity. This further supports that the most likely driver for low testosterone in patients with NAFLD remains insulin resistance, as these two diseases are inextricably linked.

In addition, this study goes beyond previous investigations in that we employed state-of-the-art techniques for NAFLD diagnosis, including the gold standard of liver biopsy as well as $^1$H-MRS. Several previous studies using ultrasound as a diagnostic marker for NAFLD showed an independent association of total and free testosterone levels with NAFLD [15–19]. However, liver ultrasound has low sensitivity for the diagnosis of NAFLD. In contrast, $^1$H-MRS results have been well correlated with liver biopsy, and have shown superior performance to hepatic ultrasound [25].

Our study had several limitations. We included only patients with type 2 diabetes, which may decrease the generalizability of our findings. However, NAFLD and type 2 diabetes are strongly linked, with up to 70% of patients with type 2 diabetes demonstrating presence of NAFLD [4]. Thus, it can be presumed that a majority of patients with type 2 diabetes would be at risk for hypogonadism related to their underlying hepatic steatosis and insulin resistance. Furthermore, there are inherent flaws in the total testosterone assay, as it is affected by sex hormone binding globulin (SHBG) and albumin levels and can be influenced by multiple underlying factors. Previous studies have indicated an association of NAFLD with total testosterone levels that is attenuated or absent when controlled for SHBG or when assessing free testosterone levels [20, 26]. Obesity and advancing age are associated with lower levels of total testosterone and SHBG [27]. Studies have shown an increase in SHBG levels and a reduced hepatic clearance of estradiol in patients with cirrhosis, producing variable effects on total testosterone levels [28]. While we did not measure SHBG levels, we assessed free testosterone, where the prevalence of hypogonadism was not different among groups. However, findings were consistent with total testosterone regarding the lack of association between testosterone levels and

histology. Future studies may benefit from using bioavailable testosterone or free testosterone by equilibrium dialysis [13, 29].

In summary, our results suggest that the prevalence of low total testosterone in T2DM is higher in patients with NAFLD compared to those without NAFLD. This is likely related to the amount of intrahepatic triglyceride content and associated insulin resistance, as opposed to liver inflammation or fibrosis. Current Endocrine Society guidelines do not recommend routine screening of men for hypogonadism, unless associated with symptoms [29]. However, our study suggests that testing for hypogonadism in patients with known NAFLD may be warranted as these patients are a very high-risk group. Further, recent studies suggest treatment of hypogonadism in men with type 2 diabetes may lead to reduced mortality [30]. Future research is needed to determine if patients with NAFLD and hypogonadism may stand to benefit from reduced mortality after treatment.

## Supporting information

**S1 Database. Database used for all data analysis.**
(CSV)

## Author Contributions

**Data curation:** Fernando Bril.

**Formal analysis:** Kristin Alexandra Dayton, Fernando Bril, Diana Barb, Jinping Lai, Srilaxmi Kalavalapalli, Kenneth Cusi.

**Funding acquisition:** Kenneth Cusi.

**Investigation:** Fernando Bril, Diana Barb, Jinping Lai, Srilaxmi Kalavalapalli, Kenneth Cusi.

**Writing – original draft:** Kristin Alexandra Dayton.

**Writing – review & editing:** Fernando Bril, Diana Barb, Jinping Lai, Srilaxmi Kalavalapalli, Kenneth Cusi.

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
