## [Decision Letter · Decision Letter 0]

26 Mar 2021

PONE-D-21-02015

Severity of Non-alcoholic Steatohepatitis is Not Linked to Testosterone Concentration in Patients with Type 2 Diabetes

PLOS ONE

Dear Dr. Cusi,

Thank you for submitting your manuscript to PLOS ONE. After careful consideration, we feel that it has merit but does not fully meet PLOS ONE’s publication criteria as it currently stands. Therefore, we invite you to submit a revised version of the manuscript that addresses the points raised during the review process.

We look forward to receiving your revised manuscript.

Kind regards,

Massimo Federici, M.D.

Academic Editor

PLOS ONE

Additional Editor Comments:

comments from reviewer 1

I read with interest the manuscript of Dayton et al regarding the influence of liver disease on testosterone in males withT2DM. In this cross-sectional study of 175 males with T2DM, NAFLD was linked to lower total testosterone in patients with T2DM, but likely given a

common soil of insulin resistance/obesity and not from the severity of liver

necroinflammation or fibrosis.

The manuscript is interesting and well written. I have some points:

- In figure 1, could you perform also a test to evaluate intra-group differences?

- In figure 4, could you perform also a test to evaluate intra-group differences for panel A and D?

- In Figure 3, were the variables logaritimically transformed? Based on Table 1, they seems not normally distribuited

- In page 12 the authors wrote: “In order to assess whether the association between total plasma testosterone and intrahepatic triglyceride content was independent of other metabolic factors, we performed a multiple linear regression analysis. Intrahepatic triglyceride content was independently associated with total testosterone levels (ß=-4.8, p=0.004). Body mass index (ß=-9.6, p<0.001) and HbA1c (ß=-19.8, p=0.02) were also independently associated with total testosterone in the multiple linear regression”. Please, could you report a table with these data?

Journal Requirements:

"Grant support for this study was provided by the Burroughs Wellcome Fund (K. C.), the

 American Diabetes Association (1-08-CR-08 [K. C.]), and a VA Merit Award (1 I01 CX000167-01 [K. C.])."

"American Diabetes Association (1-08-CR-08 [K. C.]) https://www.diabetes.org/

VA Merit Award (1 I01 CX000167-01 [K. C.]), https://www.research.va.gov/funding/

Reviewers' comments:

Reviewer's Responses to Questions

**Comments to the Author**

1. Is the manuscript technically sound, and do the data support the conclusions?

Reviewer #1: Partly

2. Has the statistical analysis been performed appropriately and rigorously? 

Reviewer #1: N/A

3. Have the authors made all data underlying the findings in their manuscript fully available?

Reviewer #1: Yes

4. Is the manuscript presented in an intelligible fashion and written in standard English?

Reviewer #1: Yes

5. Review Comments to the Author

Reviewer #1: I read with interest the manuscript of Dayton et al regarding the influence of liver disease on testosterone in males withT2DM. In this cross-sectional study of 175 males with T2DM, NAFLD was linked to lower total testosterone in patients with T2DM, but likely given a

common soil of insulin resistance/obesity and not from the severity of liver

necroinflammation or fibrosis.

The manuscript is interesting and well written. I have some points:

- In figure 1, could you perform also a test to evaluate intra-group differences?

- In figure 4, could you perform also a test to evaluate intra-group differences for panel A and D?

- In Figure 3, were the variables logaritimically transformed? Based on Table 1, they seems not normally distribuited

- In page 12 the authors wrote: “In order to assess whether the association between total plasma testosterone and intrahepatic triglyceride content was independent of other metabolic factors, we performed a multiple linear regression analysis. Intrahepatic triglyceride content was independently associated with total testosterone levels (ß=-4.8, p=0.004). Body mass index (ß=-9.6, p<0.001) and HbA1c (ß=-19.8, p=0.02) were also independently associated with total testosterone in the multiple linear regression”. Please, could you report a table with these data?

6. PLOS authors have the option to publish the peer review history of their article (what does this mean?). If published, this will include your full peer review and any attached files.

Reviewer #1: No

---

## [Author Response · Author response to Decision Letter 0]

9 Apr 2021

(letter has been attached earlier)

Massimo Federici, M.D.

Academic Editor

PLOS ONE March 26th, 2021 

RE: PLOS ONE Resubmission of manuscript PONE-D-21-02015.

Dear Dr. Federici,

We are resubmitting for publication in PLOS One our revised version of the manuscript titled entitled “Severity of NASH is Not Linked to Testosterone Concentration in Patients with Type 2 Diabetes” by Dayton et al.

We appreciate the kind words of support by the editors and reviewers, as well as their excellent suggestions to enhance our manuscript. We include below a point-by-point response to the issues raised. 

We hope that the referees and editorial board will find this improved version to meet the high standards of your prestigious Journal.

Best regards,

Kenneth Cusi, M.D., F.A.C.P., F.A.C.E. 

Professor of Medicine

 

Response to referee

Referee #1: I read with interest the manuscript of Dayton et al regarding the influence of liver disease on testosterone in males withT2DM. In this cross-sectional study of 175 males with T2DM, NAFLD was linked to lower total testosterone in patients with T2DM, but likely given a common soil of insulin resistance/obesity and not from the severity of liver necroinflammation or fibrosis.

Thank you for your support.

The manuscript is interesting and well written. I have some points:

Q1. In figure 1, could you perform also a test to evaluate intra-group differences?

After adjustment for multiple comparisons, between-group comparisons in Figure 1 are:

Group 1 vs. 2: p = 0.19

Group 2 vs. 3: p = 0.99

Group 1 vs. 3: p = 0.13

(The p we used was a p for trend so it makes sense that comparisons between groups are NS)

Data included in the legend (page 10).

Q2. In figure 4, could you perform also a test to evaluate intra-group differences for panel A and D?

Fig 4 A. After adjustment for multiple comparisons, between-group comparisons are:

Steatosis grade 1 vs. 2: p = 0.003

Steatosis grade 2 vs. 3: p = 0.99

Steatosis grade 1 vs. 3: p = 0.37

Fig4 D. After adjustment for multiple comparisons, between-stage groups comparisons are:

Stage 0 vs. 1: 0.56

Stage 1 vs. 2: 0.99

Stage 0 vs. 2: 0.65

Stage 0 vs. 3: 0.99

Stage 2 vs. 3: 0.99

Data included in the legend (page 12).

Q3. In Figure 3, were the variables logarithmically transformed? Based on Table 1, they seem not normally distributed.

We appreciate this important question from the reviewer. In Figure 3 variables were not logarithmically transformed because only liver fat and HOMA were not normally distributed based on Shapiro-Wilk normality tests. Testing these variables after logarithmic transformation, produced similar correlations to those reported in Fig. 3:

Total testosterone and liver fat: -0.28, p=0.002.

Total testosterone and HOMA: -0.30, p<0.001.

Due to the fact that the relationships were similar, for simplicity of interpretation for the readers, we decided to present the data without log transformation.

Q4. In page 12 the authors wrote: “In order to assess whether the association between total plasma testosterone and intrahepatic triglyceride content was independent of other metabolic factors, we performed a multiple linear regression analysis. Intrahepatic triglyceride content was independently associated with total testosterone levels (ß=-4.8, p=0.004). Body mass index (ß=-9.6, p<0.001) and HbA1c (ß=-19.8, p=0.02) were also independently associated with total testosterone in the multiple linear regression”. Please, could you report a table with these data?

This is a great suggestion and will add to the scientific merit of the paper. The table (below) was created and added to the manuscript as well.

 Coefficient p 95% confidence interval

Liver fat content -4.8 0.004 -7.9 -1.6

HbA1c -19.8 0.024 -37.0 -2.7

BMI -9.6 <0.001 -13.8 -5.3

ALT 0.8 0.016 0.2 1.5

Table 2: Results of multiple linear regression analysis for variables associated with total testosterone. HDL, TG, Age, FPG, AST were also tested but were not independently associated with total testosterone.

---

## [Decision Letter · Decision Letter 1]

27 Apr 2021

Severity of Non-alcoholic Steatohepatitis is not Linked to Testosterone Concentration in Patients with Type 2 Diabetes

PONE-D-21-02015R1

Dear Dr. Cusi,

We’re pleased to inform you that your manuscript has been judged scientifically suitable for publication and will be formally accepted for publication once it meets all outstanding technical requirements.

Kind regards,

Massimo Federici, M.D.

Academic Editor

PLOS ONE

Additional Editor Comments (optional):

Reviewers' comments:

Reviewer's Responses to Questions

**Comments to the Author**

1. If the authors have adequately addressed your comments raised in a previous round of review and you feel that this manuscript is now acceptable for publication, you may indicate that here to bypass the “Comments to the Author” section, enter your conflict of interest statement in the “Confidential to Editor” section, and submit your "Accept" recommendation.

Reviewer #1: All comments have been addressed

2. Is the manuscript technically sound, and do the data support the conclusions?

Reviewer #1: Yes

3. Has the statistical analysis been performed appropriately and rigorously? 

Reviewer #1: Yes

4. Have the authors made all data underlying the findings in their manuscript fully available?

Reviewer #1: Yes

5. Is the manuscript presented in an intelligible fashion and written in standard English?

Reviewer #1: Yes

6. Review Comments to the Author

Reviewer #1: Thank you very much for addressing my points. The maunuscript has been improved. No further comment.

7. PLOS authors have the option to publish the peer review history of their article (what does this mean?). If published, this will include your full peer review and any attached files.

Reviewer #1: No

---

## [Editor Report · Acceptance letter]

21 May 2021

PONE-D-21-02015R1 

Severity of Non-alcoholic Steatohepatitis is not Linked to Testosterone Concentration in Patients with Type 2 Diabetes 

Dear Dr. Cusi:

I'm pleased to inform you that your manuscript has been deemed suitable for publication in PLOS ONE. Congratulations! Your manuscript is now with our production department. 

Kind regards, 

on behalf of

Prof. Massimo Federici 

Academic Editor

PLOS ONE